# Role of Platelet-Activating Factor in the Pathogenesis of Chronic Spontaneous Urticaria

**DOI:** 10.3390/ijms252212143

**Published:** 2024-11-12

**Authors:** Bo Youn Choi, Young-Min Ye

**Affiliations:** Department of Allergy and Clinical Immunology, Ajou University School of Medicine, Suwon 16499, Republic of Korea

**Keywords:** platelet-activating factor, chronic spontaneous urticaria, mast cells, degranulation

## Abstract

Chronic spontaneous urticaria (CSU) is a debilitating condition characterized by mast cell activation. Platelet-activating factor (PAF) is produced by various immune cells, including mast cells, basophils, lymphocytes, and eosinophils, which play crucial roles in CSU pathogenesis. It induces mast cell degranulation, increases vascular permeability, and promotes the chemotaxis of inflammatory cells. These effects result in the release of inflammatory mediators, the development of edema, and the persistence of inflammation, which are key features of CSU. Notably, elevated PAF levels have been linked to heightened disease activity and resistance to antihistamine treatment in CSU patients. Despite these findings, the precise role of PAF in CSU pathogenesis remains unclear. Rupatadine, an antihistamine, and heat shock protein 10, a natural anti-inflammatory peptide that selectively inhibits PAF-induced mast cell degranulation, have demonstrated anti-PAF activity. Furthermore, with the molecular structure of the PAF receptor now identified, several experimental PAF receptor antagonists have been synthesized. However, there remains a significant need for the development of therapeutic options targeting PAF in CSU management.

## 1. Introduction

Chronic spontaneous urticaria (CSU) is characterized by itchy wheals and/or angioedema persisting for at least 6 weeks, with no identifiable extrinsic triggers [1]. With a global prevalence of approximately 1.4% and a long-lasting, often relapsing disease course, CSU substantially impairs the quality of life and imposes a considerable socioeconomic burden [2,3,4]. Current consensus suggests that CSU is primarily driven by mast cell activation and degranulation [2]. As a chronic inflammatory disorder, it involves the infiltration of various immune cells, including mast cells, basophils, eosinophils, and a mixed population of T cells, along with increased expression of adhesion molecules and cytokines [5].

Two distinct endotypes have been identified, representing the most plausible intrinsic triggers for prolonged mast cell activation and degranulation. Type I autoimmunity involves autoreactive IgE antibodies, which degranulate mast cells by crosslinking autoantigens and binding to their high-affinity receptors [6,7,8]. In type IIb autoimmune CSU, IgG autoantibodies targeting IgE and FcεRI are involved. However, these endotypes do not fully account for the persistent mast cell activation seen in CSU patients, as they are identified in less than half of the cases. Increasing evidence highlights additional mechanisms, including platelet-activating factor receptor (PAFR), Mas-related G-protein coupled receptor X2, basophil alarmins, and other signaling pathways, contributing to mast cell degranulation [9,10,11,12,13,14,15,16,17,18,19,20,21,22] (Figure 1).

Considering the complexity of CSU pathogenesis, disease characteristics and treatment responses can vary significantly among patients. These differences are influenced by the primary triggers of mast cell degranulation, the dominant immune cells infiltrating the tissue, and the dysregulated cytokines involved in chronic inflammation. Identifying these triggers and inflammatory patterns in individual patients is crucial for advancing precision medicine in the treatment of CSU. In this review, we explore the role of platelet-activating factor (PAF) in the pathogenesis of CSU.

## 2. PAF and PAFR

### 2.1. PAF

PAF is a potent phospholipid mediator involved in various physiological and pathological processes, including platelet aggregation, allergies, asthma, inflammation, cancer, atherosclerosis, and neurological diseases. PAF is produced by various cells in response to specific stimuli, and it is synthesized via two distinct pathways: the remodeling pathway and the de novo pathway [23] (Figure 2). In the remodeling pathway, phospholipase A2 and lyso-PAF acetyltransferase serve as key regulatory enzymes, whereas dithiothreitol (DTT)-sensitive CDP-choline phosphotransferase is the main regulatory enzyme in de novo synthesis [24,25]. In both acute and chronic inflammation, PAF is predominantly produced via the remodeling pathway, whereas PAF needed for normal cellular functions is synthesized through the de novo pathway. Due to its proinflammatory effects, PAF is widely used as a biomarker for various chronic inflammatory diseases [26,27]. PAF, which is metabolized by PAF acetylhydrolase (PAF-AH), has a short half-life of 3–13 min [28]. As a result, increased PAF-AH activity leads to a reduction in PAF levels.

PAF can activate platelets, playing a critical role in blood coagulation and hemostasis. It is involved in atherosclerosis and directly affects heart function, contributing to cardiovascular diseases. Additionally, as an inflammatory mediator, it acts on various inflammatory cells, including neutrophils, monocytes, macrophages, and mast cells. It promotes leukocyte chemotaxis and induces the release of cytokines and chemokines by activating these cells while also increasing endothelial cell permeability. It also exhibits neurotrophic effects in the brain and plays a role in reproductive processes [29,30,31]. Furthermore, it can induce the release of histamine and other inflammatory mediators through the activation and degranulation of mast cells and basophils [9,11,32,33].

### 2.2. PAFR

PAF has both autocrine and paracrine functions, exerted by binding to the G-protein-coupled receptor (PAF-R), which is expressed on both plasma and nuclear membranes. The *PAFR* gene, located on chromosome 1, contains two unique promoters and produces two distinct PAFR mRNA transcripts in human tissue [34,35,36,37]. PAFR transcript 1 is expressed in hematopoietic cells, whereas transcript 2 is found in various organs, such as the heart, lungs, spleen, and kidneys [34,36]. Transcript 1 is primarily involved in pathological processes through the PKC and NF-κB pathways, whereas transcript 2 is regulated by hormones and cytokines, including estrogen, thyroid hormone T3, retinoic acid, and TGF-β [35]. Additionally, the transcriptional activation of transcript 1 is regulated by PAF [38].

Once activated by an agonist, PAFR couples with Gq, Gi, G12/13, and β-arrestin to activate downstream signaling pathways, including PKC, cAMP, and PAFR internalization. Following the structural elucidation of PAFR, ligand-induced dimerization or oligomerization of PAFR, leading to biased signaling, may occur [39,40,41,42,43] (Figure 3). Studies on a natural genetic variant have shown that this variant is more likely to induce dimerization, thereby amplifying both physiological and pathophysiological effects. Additionally, increased PAFR expression promotes the formation of dimers and oligomers, further enhancing PAF signaling [41].

The correlation between CSU and the PAFR genetic variant remains unknown. However, a study using skin samples from CSU patients and healthy controls found that PAFR mRNA expression was significantly higher in CSU patients. Furthermore, endothelial cell PAFR expression was observed exclusively in CSU patients [13].

## 3. Clinical Implications of PAF in CSU

PAF is released by various cells involved in the pathogenesis of CSU (Table 1). 

All PAF-producing cells, including mast cells, basophils, eosinophils, neutrophils, macrophages, monocytes, endothelial cells, and keratinocytes, express PAFR on their membranes. Consequently, PAF functions in both autocrine and paracrine manners. Despite the crucial role of PAF in CSU pathogenesis, both independently and through its interactions with various immune cells, PAF has not received significant attention compared to other mediators, such as histamine. PAF can induce the release of histamine, IL-4, IL-5, IL-6, and IL-13 from mast cells and basophils—key effector cells in CSU. Additionally, it enhances the Th2 response and recruits other inflammatory cells into the tissue, highlighting its critical role in the pathogenesis of CSU. This review summarizes recent findings on the clinical implications of PAF in CSU, with a focus on how it drives mast cell activation, degranulation, and proliferation, which are key processes in CSU pathogenesis (Figure 4). In addition, we examine the impact of PAF on vascular permeability, its contribution to endothelial damage, and its role in immunomodulation, all of which have significant implications for disease progression and potential therapeutic strategies.

### 3.1. Elevated PAF Levels in Patients with CSU

In a previous study, we demonstrated that serum levels of PAF were elevated in CSU patients and were even higher in those who did not respond to H1-antihistamine treatment [12]. Additionally, PAF-AH levels were reduced in CSU patients compared to healthy controls. Furthermore, a lower PAF-AH/PAF ratio was observed in H1-antihistamine non-responders compared to those who responded well to H1-antihistamine treatment [12,13]. PAF-AH levels positively correlated with BMI and complemented C3 and C4, while negatively correlating with urticaria duration. However, neither PAF nor PAF-AH levels were correlated with urticaria activity scores. While we observed a significant increase in serum levels of PAF in a large cohort of 283 CSU patients compared to 111 healthy controls, two other studies with smaller sample sizes (<30 subjects in each group) reported higher PAF levels in CSU patients, but without statistical significance [12,57]. The high inter-individual variability in PAF levels, observed in both CSU patients and healthy controls, could explain these differences. This variability, likely influenced by the short half-life of PAF, poses challenges in consistently detecting differences, particularly in smaller cohorts.

### 3.2. PAF as a Mast Cell Activator and Secretagogue

PAF induces wheals, flare, and pruritus in a dose-dependent manner when injected intradermally into human skin, with effective doses ranging from 10 to 100 ng [58,59]. Sciberras et al. observed an immediate (<5 min) increase in plasma levels of histamine following intradermal PAF injections in both atopic and non-atopic individuals, with atopic subjects initially having double the baseline histamine levels [60].

Furthermore, antihistamine treatment and prior histamine depletion using compound 48/80 significantly reduced PAF-induced pruritus, flares, and wheals. These findings suggest that histamine is a key mediator in PAF-induced symptoms and that non-IgE-mediated histamine release can occur during PAF-induced reactions [58,59,60]. However, Thomas et al. found that isolated human skin mast cells are unable to degranulate in response to PAF, whereas PAF induces histamine release from human lung mast cells and mast cells derived from peripheral blood (PB) [61]. In that study, PAF at concentrations of 10^−9^ and 10^−8^ mol/L exhibited an additive effect on FcεRI-mediated histamine release, while concentrations higher than 10^−8^ mol/L resulted in a diminished effect compared to the expected additive response. They also identified that the degranulation of lung and PB-derived mast cells in response to PAF was mediated through the activation of the PAFR-coupled G protein Gαi, which is sensitive to pertussis toxin. This pathway is shared by both FcεRI- and PAFR-mediated mast cell activation [61]. Further in vitro experiments with the LAD2 human mast cell line and primary human lung mast cells have demonstrated that PAF induces mast cell degranulation, as evidenced by beta-hexosaminidase and histamine assays [14,62].

PAF also induces histamine release from human basophils, which is optimal after incubation in Ca^2+^ for 2–5 min. This effect is further enhanced by IL-3 and GM-CSF [63]. Basophils show partial desensitization to PAF (100 nM to 1 µM) after a 2 min preincubation without Ca^2+^ but remain responsive to anti-IgE stimulation [64]. Notably, there is no significant difference in histamine and LTC4 release between allergic and non-allergic donors upon PAF stimulation [64]. The release mechanism triggered by PAF is independent of IgE-mediated pathways and other stimuli such as C5a and FMLP and is specific to PAF itself.

### 3.3. PAF-Induced Vascular Permeability and Endothelial Damage

When PAF is injected into the skin of experimental animals, it causes an immediate increase in vascular permeability and subsequent plasma extravasation [65]. In human skin, intradermal administration of PAF induces acute wheal and erythema, accompanied by neutrophil and monocyte accumulation. Histopathologic studies have revealed that infiltration by neutrophils and eosinophils can occur as early as 30 min and persist for up to 4 h after PAF administration [66]. In CSU patients, PAFR expression is significantly elevated in lesional skin at both mRNA and protein levels, with PAFR localized primarily in the epidermis and endothelium [13].

Increased vascular permeability is a key characteristic of urticarial wheals. Neutrophils pre-treated with PAF display enhanced responses to agonists such as FMLP and PMA, releasing superoxide and elastase, which increase endothelial adhesion and promote cell damage [67]. Endothelial cells themselves produce PAF in response to proinflammatory and vasoactive ligands, including histamine, bradykinin, leukotrienes, thrombin, and cytokines such as TNF or IL-lα [68,69,70]. PAF also triggers intracellular calcium elevation and cytoskeletal protein reorganization in endothelial cells, which together increase vascular permeability after PAF exposure [71]. The resulting endothelial damage facilitates platelet and mast cell interactions, with platelet-derived PAF contributing to mast cell degranulation in CSU lesions.

Furthermore, platelet-derived PAF is sufficient to induce shock, characterized by vascular leakage, tissue inflammation, and decreased core temperature, as shown in a systemic platelet activation mouse model [11]. It has been reported that the platelet count in peripheral blood is significantly higher in patients with moderate to severe CSU than in healthy individuals and those with mild CSU [72]. With a significant correlation to C-reactive protein (CRP), a well-known biomarker of CSU in terms of disease severity and poor response to antihistamine treatment, the increased platelet count suggests its potential role in CSU pathogenesis [73]. Therefore, the role of PAF in the pathogenesis of CSU is not limited to mast cell degranulation but also involves the priming of neutrophils, leading to endothelial cell lysis. Both effects of PAF have been found to be alleviated by a specific PAFR antagonist [11,67].

PAFR expression on mast cells is tissue-specific. Lung mast cells and PB-derived mast cells express PAFR, whereas skin mast cells do not [61]. Isolated human skin mast cells do not degranulate in response to PAF [74]. However, in a microdialysis analysis of intact human skin tissue, PAF-induced histamine release was suggested to be mediated by neurogenic activation, which was reduced by nerve blocking [59].

The low expression of PAFR on skin mast cells suggests that direct activation of these cells by PAF may be limited. However, PAF can also influence skin mast cells indirectly by activating other immune cells, such as eosinophils, neutrophils, or macrophages, which subsequently release mediators that can activate mast cells and may even enhance PAFR expression.

In addition, while the main subpopulation of human skin mast cells (MC_TC_) does not significantly differ between lesional skin, non-lesional CSU skin, and healthy control skin [75], the number of tryptase-positive and chymase-negative mast cells (MC_T_), the major population in peripheral blood, is significantly increased in CSU lesional skin compared to non-lesional CSU and healthy individuals [76]. This indicates a relatively increased presence of PAFR-expressing MC_T_ in the lesional skin of CSU patients. Taken together, these results indicate that circulating mast cells can migrate to the skin through damaged endothelium, especially in active lesions, and phenotypic changes can be induced by the characteristic inflammation of CSU skin.

### 3.4. PAF as an Immune Modulator in CSU

T-lymphocytes were identified as the main component of the inflammatory infiltrate expressing PAFR in the lesional skin of CSU patients [13]. PAF induces IL-4 production in peripheral blood mononuclear cells (PBMCs) from both healthy donors and CSU patients [14,77]. PAF-induced IL-4 secretion activates Th2 cells, promotes antibody production, enhances LDL oxidation, and increases IL-6 secretion, contributing to both chronic and acute inflammatory processes [77].

Given that both PAF-AH and complement secretion are stimulated by proinflammatory cytokines such as IL-6 and TNF-α, recombinant PAF-AH has shown potential in reducing inflammation in various experimental models [78]. In a recent study, we found that PAF-AH expression is downregulated by microRNA-101-5p, which is released from endothelial cells and human keratinocytes in response to IL-4 stimulation [14]. Huang et al. demonstrated that PAF can stimulate the production of IL-4 and IL-6 in PBMCs from healthy controls, an effect that is diminished by the specific PAFR antagonist WEB 2170 [77]. Notably, in one study, a significant increase in IL-4+ cells was observed in lesional CSU skin compared to controls [79], and Th2-initiating cytokines, including IL-33, IL-25, and thymic stromal lymphopoietin (TSLP), were elevated in lesional CSU skin compared to non-lesional CSU skin and controls. Therefore, the enhanced secretion of IL-4 and IL-6 from activated T cells in CSU skin may upregulate microRNA-101-5p, subsequently reducing PAF-AH levels and prolonging PAF activity. In summary, upregulating PAF-AH—by inhibiting proinflammatory cytokines and blocking IL-4 action—holds therapeutic potential to reduce PAF-mediated mast cell activation and degranulation.

Basophil infiltration has been observed in CSU, and basopenia (reduced basophils in peripheral blood) is significantly associated with higher urticaria activity scores and decreased responsiveness to antihistamine and anti-IgE treatments [2,18,73,80]. The exact mechanism of basophil recruitment to the skin remains unclear. It is assumed that basophils are attracted by various mediators in the skin, including chemokines such as CCL2, CCL5, CCL11, CXCL12, and prostaglandin D2, with basophils expressing corresponding receptors such as CCR4, CCR3, CXCR4, and CRTH2. PAF also induces epigenetic modifications in human mast cells [81]. Damiani et al. investigated the effects of UV-induced keratinocyte-derived PAF on mast cells and found that CXCR4 expression was upregulated in human mast cell lines (HMC-1 cells) 6–24 h after treatment with carbamyl-PAF at a concentration of 10 µM [81]. Simultaneously, PAF induced a reduction in DNMT1 in HMC-1 cells for up to 48 h after exposure, suggesting increased demethylation of the CXCR4 promoter, leading to increased CXCR4 expression. These epigenetic modifications induced by PAF can occur in basophils, enhancing their migration to CSU lesions and contributing to basopenia. Additionally, TSLP [82] and IL-31 [83], which are elevated in CSU patients, have been shown to induce chemotaxis in basophils in vitro [84].

PAF has been shown to stimulate cyclic AMP (cAMP) generation in normal human B lymphocytes in a time- and dose-dependent manner [85]. In that study, when PAF was added at a concentration of 10^−6^ M to IL-2 or IL-4, which alone do not elicit a cAMP response in B cells, it enhanced cAMP production beyond levels induced by PAF alone. However, PAF has differential effects on IgM synthesis in B cells stimulated by IL-2 and IL-4. While in that study, PAF downregulated IL-4-induced IgM secretion, it increased IL-2-induced IgM secretion. These effects were eliminated with the use of CV-3988, a PAFR antagonist [85]. The differential response of PAF on the effects of IL-2 and IL-4 in human B cells may be due to the fact that IL-2 responses are protein kinase C (PKC)-dependent while IL-4 responses are PKC-independent [86]. Therefore, PAF acts as an immunomodulator in B cells, and its effects can vary depending on the inflammatory milieu in patients with CSU.

PAFR expression has been shown to decrease 48 h after LPS stimulation in human dendritic cells (DCs) [87]. Koga et al. investigated the effect of PAF on DC phenotype and antigen-presenting function using a murine model [88]. When DCs exposed to LPS produce PAF, it binds to PAFR and enhances the synthesis of IL-10 and PGE2, inducing a regulatory phenotype in DCs. The immunogenicity of the skin is closely linked to the number of resident DCs, which are crucial for activating naïve T cells [89].

PAF induces Th17 cell differentiation via interactions with Langerhans cells [90]. It triggers the immediate expression of IL-1ß, IL-6, and IL-23 in human Langerhans cells and keratinocytes; cytokines that are essential for Th17 development. Treatment with PAFR antagonists, along with anti-IL-6R and anti-IL23, block the expression of the Th17-specific transcriptional regulator RORγτ in response to PAF. In previous studies, elevated plasma levels of IL-17 and IL-23 have been observed in patients with CSU, correlating with urticaria severity [83,91]. Additionally, IL-23 levels are significantly higher in patients with positive autologous serum skin test (ASST) results compared to those with a negative ASST [91]. Consistent with increases in IL-17 and IL-23 observed in various autoimmune diseases, a negative ASST in CSU has been suggested to be a reliable marker to exclude the presence of functional circulating autoantibodies detected by the basophil histamine release assay. The potentiation of autoimmune characteristics, related to an increase in Th17 response upon PAFR activation, may explain the elevated PAF levels in antihistamine-refractory CSU patients.

### 3.5. PAF as a Trigger of Neural Activation in CSU

PAF plays a role in neural activation in CSU. It is known that neuronal cells can produce PAF, and functional PAFR has been identified on neuronal membranes, suggesting a direct pathway for PAF’s effects on nerve cells [92]. Fjellner et al. demonstrated that PAF-induced pruritus in human skin is mediated primarily by an indirect, histamine-dependent mechanism, which indicates that PAF may trigger itching in CSU through the activation of histamine-releasing cells [58].

Moreover, Marotta et al. found that local injection of PAF into rat paws induces spontaneous nociception and mechanical hypersensitivity, suggesting that PAF sensitizes peripheral nerves to painful stimuli. PAF-induced changes, such as IL-1ß activation, TRPV1 activation, and mast cell degranulation—features observed in CSU lesions—may be contributing mechanisms [92].

Pruritus is a hallmark symptom of both atopic dermatitis (AD) and CSU, driven in part by the activation of sensory neurons. In AD, itch signaling involves sensory neurons that express receptors for TSLP, IL-4, and IL-13—cytokines also implicated in CSU pathogenesis. A double-blind, randomized, within-patient study evaluating the efficacy of topical application of a PAF antagonist in patients with AD demonstrated a significant reduction in pruritus severity during the first two weeks, despite no significant changes in erythema, scaling, induration and exudation [93]. The investigators suggested that topical PAF antagonist may inhibit inflammatory pathways associated with arachidonic acid by reducing leukotriene B4 and thromboxane A2. Additionally, prostaglandins E1 and E2 are known to lower the itch threshold for histamine. Therefore, by reducing prostaglandin levels, PAF antagonists may help alleviate histamine-induced itching.

## 4. Therapeutic Potential of Targeting PAF in CSU

To date, evidence supporting the clinical efficacy of selective PAF antagonists in human CSU and allergic diseases is limited. Rupatadine, an H1-antihistamine, exhibits antagonistic effects on the PAF receptor both in vitro and in vivo [94]. Munoz-Cano et al. [62] demonstrated that rupatadine inhibits PAF-induced degranulation at a concentration range of 1 µM to 10 µM in both LAD2 cells and human lung tissue mast cells. Conversely, desloratadine shows no inhibitory effect, and levocetirizine inhibits PAF-induced degranulation in LAD2 cells. A recent randomized controlled trial compared the efficacy of a combination treatment involving antiplatelet drugs, such as cilostazol and dipyridamole, with the up-dosing of antihistamines [95]. The study found that the combination treatment was more effective in reducing the severity of urticaria symptoms. This suggests that antiplatelet drugs may offer therapeutic potential for a specific subset of CSU patients, particularly those with signs of platelet activation, such as elevated D-dimer levels, who do not achieve adequate symptom control with antihistamine treatment alone.

In light of the role of PAF in inducing IL-17 and IL-23 production, and the observation of elevated levels of these cytokines in severe CSU, treatments targeting IL-17 and IL-23 have been explored. Secukinumab, an anti-IL-17 monoclonal antibody, was proven to be effective in a study of eight patients with severe CSU refractory to omalizumab, showing an 82% reduction in UAS7 scores after 3 months of treatment [96]. In addition, a case series involving three patients with antihistamine- and omalizumab-refractory CSU revealed that anti-IL-23 treatment with tildrakizumab effectively improved urticarial symptoms, achieving disease control and enhancing quality of life [97].

Dupilumab, a fully human monoclonal antibody that blocks both IL-4 and IL-13, has been evaluated in patients with symptomatic CSU receiving antihistamine treatment [98]. In one study, it led to reductions in itchy, wheal, and urticaria activity scores over 24 weeks in patients with antihistamine-refractory CSU, as well as in those with an incomplete response to omalizumab, although changes in the omalizumab incomplete responder arm were not statistically significant. Given the interaction between PAF and IL-4 in PBMCs and keratinocytes, blocking IL-4 signaling may help alleviate the PAF-induced mast cell degranulation, reduce vascular permeability, and decrease neuro-immune activation in CSU pathogenesis, particularly in patients with antihistamine refractory CSU.

We have identified heat shock protein 10 (HSP10) as a natural anti-PAF molecule [14]. HSP10 shows a significant correlation with PAF-AH, indicating decreased PAF levels. Higher initial PAF-AH levels are significantly associated with better urticaria control after 3 months of antihistamine treatment. Furthermore, HSP10 prevents mast cell degranulation in response to PAF. Notably, the inhibitory effect of HSP10 on mast cell degranulation is solely dependent on PAF stimulation and is not mediated by FcεR1 or MRGX2. Additionally, HSP10 reduces PAF-induced IL-4 production, further contributing to its anti-PAF effects.

In one study, a single 40 mg dose of rupatadine in healthy volunteers reduced PAF-induced flares by 87% and PAF-induced platelet aggregation by 82% within 2 h [99]. Although the inhibitory effect of rupatadine on PAF-induced mast cell degranulation is believed to be mediated through the PAF receptor, the exact mechanism remains to be elucidated.

Considering the higher prevalence of anti-HSP10 IgG autoantibodies in CSU patients, modulating autoimmune reactions and increasing HSP10 production represent promising therapeutic options for those with urticaria symptoms linked to PAF activity. However, because measuring serum levels of PAF and PAF-AH remains largely confined to research settings, reliable surrogate markers or clinical characteristics that reflect PAF involvement in CSU pathogenesis are needed. These efforts could lead to the development of more effective anti-PAF therapies.

In summary, PAF plays a multifaceted role in the pathogenesis of CSU, distinct from the IgE-mediated mechanisms. PAF contributes to CSU through non-IgE-mediated pathways, including the activation of inflammatory cells, modulation of vascular permeability, and both direct and indirect mast cell degranulation, as well as stimulation of sensory neurons. Unlike IgE-mediated hypersensitivity, which is often acute and allergen-specific, PAF-mediated effects in CSU suggest a broader, chronic inflammatory process that perpetuates itch and inflammation without the need for specific allergens.

The ability of PAF to activate various immune and non-immune cells, including mast cells, basophils, neutrophils, endothelial cells, and sensory neurons, underscores its role as a central mediator in sustaining the chronic inflammation and pruritus characteristics of CSU. Given these pathways, targeting PAF or PAFR could offer therapeutic benefits. Future research to further elucidate PAF signaling pathways in CSU could provide valuable insights and guide the development of novel treatment strategies.

## Figures and Tables

**Figure 1 ijms-25-12143-f001:**
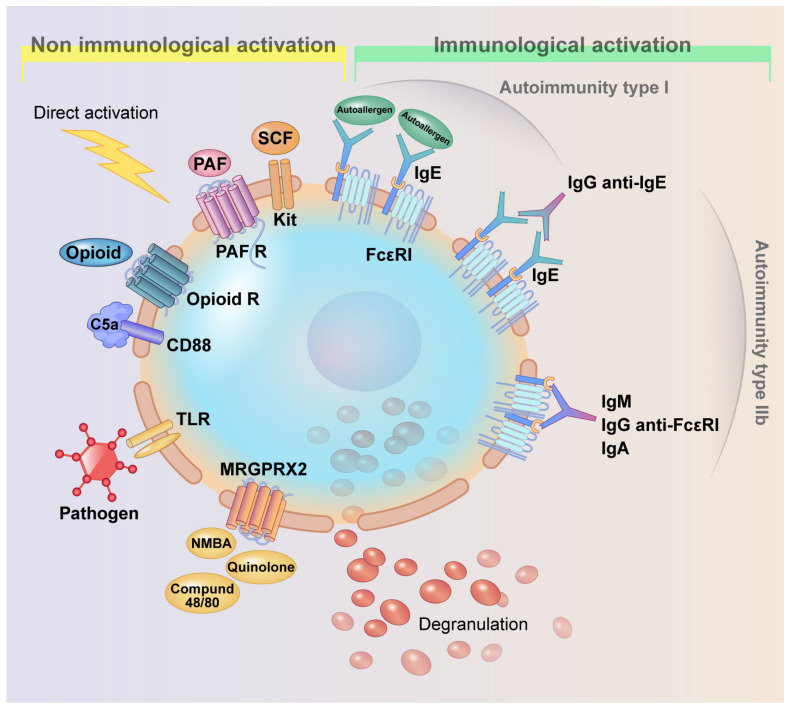
Mast cell degranulation. Modified from a previous study [22].

**Figure 2 ijms-25-12143-f002:**
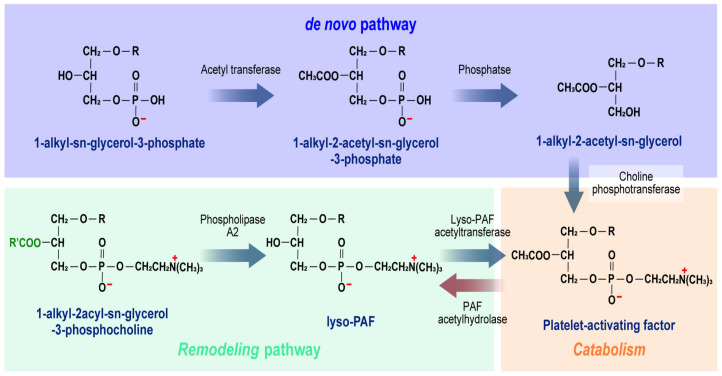
Pathways of PAF synthesis and metabolism.

**Figure 3 ijms-25-12143-f003:**
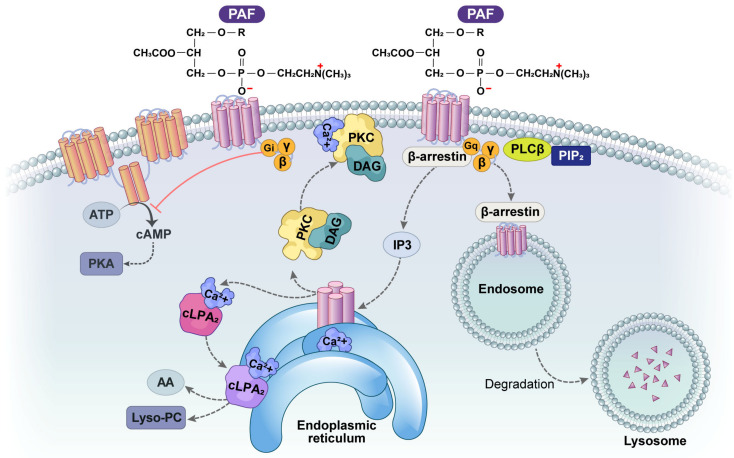
PAF-PAFR signaling.

**Figure 4 ijms-25-12143-f004:**
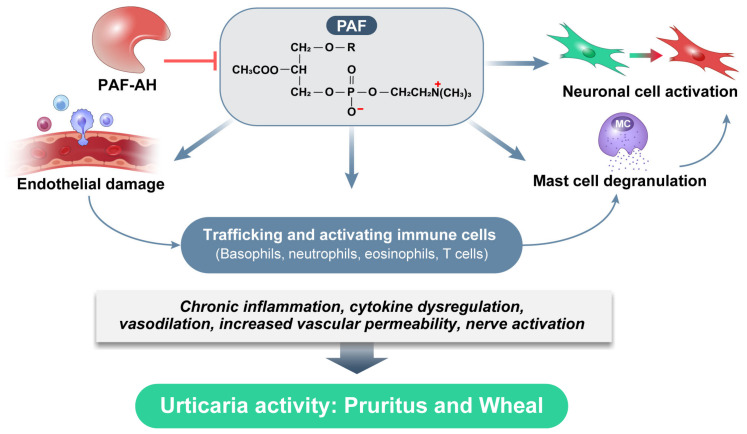
Role of PAF in the pathogenesis of chronic spontaneous urticaria.

**Table 1 ijms-25-12143-t001:** Immune cells that produce PAF, express PAFR, and release cytokines upon PAF-induced activation.

Cells	PAF Release	PAFR Expression	Released Cytokines	References
Mast cells	o	o	Histamines, TNFα, IL-4, IL-5, and IL-6. Increase vascular permeability and recruit other inflammatory cells, enhancing Th2 response	[14,33,44,45]
Basophils	o	o	Histamines, IL-4, and IL-13. Enhance Th2 response and maintain chronic inflammation	[44,45,46]
Keratinocytes	o	o	Arachidonic acid, PGE2, Cox2, IL-6, IL-8, and ROS	[47,48]
Neutrophils	o	o	IL-8, TNFα, and IL-1β. Recruit immune cells and enhance immune response	[49,50]
Macrophages	o	o	TNFα, IL-6, and IL-1β. Promote immune cell infiltration	[51]
Eosinophils	o	o	IL-4, IL-5, IL-13, and ROS. Exacerbate the allergic response	[52]
Monocytes	o	o	IL-1β, TNFα, and IL-6. Promote inflammatory responses	[53,54]
Endothelial cells	o	o	IL-8, monocyte chemoattractant protein-1 (MCP-1), and vascular endothelial growth factor (VEGF). Promote leukocyte extravasation and angiogenesis	[46,55,56]

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
