# Peer review of "Role of Platelet-Activating Factor in the Pathogenesis of Chronic Spontaneous Urticaria"

_ijms, 2024, doi:10.3390/ijms252212143_

Round 1

Reviewer 1 Report

Comments and Suggestions for Authors

This is a cmprehensive review summarizes the important rle of PAF in CSU. It describes well the pathogenic role of PAF in iducing angiogenesis and endothelial activation. It also describes the relation of PAF to CSU, specificaly mentioning the benefit of Rupatadine in CSU.

Some comments:

1. Section: 2 PAF and 3 PAFR should be shortened and combined into one section. First two paragraphs of 2PAF are too detailed and of no relevance to CSU
2. Section 4: 4.3 "PAF induced vascular permeability is too long and prevents readers from the main point of PAF and CSU. Therefore it should be more focused and shortened.
In general this is a good review which put light onto the importance of PAF in CSU

Author Response

Comment 1. Section: 2 PAF and 3 PAFR should be shortened and combined into one section. First two paragraphs of 2 PAF are too detailed and of no relevance to CSU

Response 1. In accordance with your recommendation, we have revised and combined the sections on PAF and PAFR, shortening the content to enhance relevance. Additionally, the first two paragraphs of Section 2 (PAF) have been condensed to remove excessive detail not directly pertinent to CSU.

Comment 2. Section 4: 4.3 "PAF induced vascular permeability is too long and prevents readers from the main point of PAF and CSU. Therefore it should be more focused and shortened.
In general this is a good review which put light onto the importance of PAF in CSU

Response 2. Thank you for your thoughtful recommendation. We have revised Section 4.3 to make it more focused and concise, highlighting the key points on PAF in CSU at the last of the manuscript. We appreciate your positive feedback on the review’s emphasis on the importance of PAF in CSU.

Reviewer 2 Report

Comments and Suggestions for Authors

The review article entitled "Role of Platelet-Activating Factor in the Pathogenesis of Chronic Spontaneous Urticaria" is informative, but there are some concerns that need to be addressed as follows.

Major Concerns

(1)   Besides chronic spontaneous urticaria (CSU), what is the role of platelet-activating factor (PAF) in the pathogenesis of acute urticaria which is the most cases of urticaria? Likewise, what about the case of chronic inducible urticaria?

(2)   In the case of atopic dermatitis (AD), sensory neurons expressing receptors for TSLP, IL-4 and IL-31 in basophils are activated and drive itch sensation. Given that the most important hallmark of urticaria progression is the development of pruritus as well as in the case of AD, the authors should discuss how does PAF activate sensory neurons and transmit itch signals that drive itch sensation and scratching behaviors.

(3)   To clarify the ambiguous role of PAF in CSU pathogenesis in comparison to IgE-mediated Type I hypersensitivity reactions, we encourage authors to include the closing paragraph and summarize the main points discussed in the paper and provide a final thought or conclusion.

Author Response

(1)   Besides chronic spontaneous urticaria (CSU), what is the role of platelet-activating factor (PAF) in the pathogenesis of acute urticaria which is the most cases of urticaria? Likewise, what about the case of chronic inducible urticaria?

Response 1: To date, there have been no studies specifically measuring PAF levels in cases of acute urticaria or chronic inducible urticaria. However, a recent paper on anaphylaxis, where acute urticaria is one of the characteristic manifestations, has documented elevated PAF levels, suggesting a potential role of PAF in these conditions.

(2)   In the case of atopic dermatitis (AD), sensory neurons expressing receptors for TSLP, IL-4 and IL-31 in basophils are activated and drive itch sensation. Given that the most important hallmark of urticaria progression is the development of pruritus as well as in the case of AD, the authors should discuss how does PAF activate sensory neurons and transmit itch signals that drive itch sensation and scratching behaviors.

Response 2. Thank you for this valuable suggestion. We have added a paragraph in Section 4.4 discussing pruritus as a key symptom shared by both atopic dermatitis (AD) and chronic spontaneous urticaria (CSU), highlighting the role of sensory neurons in itch signaling. In AD, cytokines such as TSLP, IL-4, and IL-13, which are also implicated in CSU pathogenesis, activate sensory neurons to drive itch. We also referenced a study showing that topical application of a PAF antagonist significantly reduced pruritus severity in AD, likely by inhibiting inflammatory pathways linked to arachidonic acid and lowering levels of leukotriene B4 and thromboxane A2. Since prostaglandins E1 and E2 can lower the histamine itch threshold, reducing prostaglandin levels through PAF antagonists may help alleviate histamine-induced itching. 

(3)   To clarify the ambiguous role of PAF in CSU pathogenesis in comparison to IgE-mediated Type I hypersensitivity reactions, we encourage authors to include the closing paragraph and summarize the main points discussed in the paper and provide a final thought or conclusion.

Response 3. 

Thank you for your suggestion. We have added a concluding paragraph to clarify the role of PAF in CSU pathogenesis in contrast to IgE-mediated Type I hypersensitivity reactions. The conclusion reads as follows: 'In summary, PAF plays a multifaceted role in the pathogenesis of CSU, distinct from IgE-mediated mechanisms. PAF contributes to CSU through non-IgE-mediated pathways, including the activation of inflammatory cells, modulation of vascular permeability, and both direct and indirect mast cell degranulation, as well as stimulation of sensory neurons. Unlike IgE-mediated hypersensitivity, which is often acute and allergen-specific, PAF-mediated effects in CSU suggest a broader, chronic inflammatory process that perpetuates itch and inflammation without specific allergens. The ability of PAF to activate various immune and non-immune cells, including mast cells, basophils, neutrophils, endothelial cells, and sensory neurons, underscores its role as a central mediator in sustaining the chronic inflammation and pruritus characteristic of CSU. Given these pathways, targeting PAF or PAFR could offer therapeutic benefits. Future research to further elucidate PAF signaling pathways in CSU could provide valuable insights and guide the development of novel treatment strategies.' This addition provides a comprehensive summary of the main points discussed in the paper and offers a final perspective on PAF’s role and therapeutic potential in CSU."

Round 2

Reviewer 2 Report

Comments and Suggestions for Authors

The paper is suitable for publication.